# Late Pleistocene South American megafaunal extinctions associated with rise of Fishtail points and human population

Luciano Prates [1,2,4 ✉] & S. Ivan Perez [1,3,4 ✉]

In the 1970s, Paul Martin proposed that big game hunters armed with fluted projectile points colonized the Americas and drove the extinction of megafauna. Around fifty years later, the central role of humans in the extinctions is still strongly debated in North American archaeology, but little considered in South America. Here we analyze the temporal dynamic and spatial distribution of South American megafauna and fluted (Fishtail) projectile points to evaluate the role of humans in Pleistocene extinctions. We observe a strong relationship between the temporal density and spatial distribution of megafaunal species stratigraphically associated with humans and Fishtail projectile points, as well as with the fluctuations in human demography. On this basis we propose that the direct effect of human predation was the main factor driving the megafaunal decline, with other secondary, but necessary, co-occurring factors for the collapse of the megafaunal community.

[1] Consejo Nacional de Investigaciones Científicas y Técnicas, Buenos Aires, Argentina. [2] División Arqueología, Facultad de Ciencias Naturales y Museo, Universidad Nacional de La Plata, La Plata, Argentina. [3] División Antropología, Facultad de Ciencias Naturales y Museo, Universidad Nacional de La Plata, La Plata, Argentina. [4] These authors contributed equally: Luciano Prates, S. Ivan Perez. ✉email: lprates@fcnym.unlp.edu.ar; ivanperezmorea@gmail.com

A great number of megafaunal species became extinct all over the planet—except for Africa—during the end of Pleistocene as their ecological niches experienced significant changes. The impact of diverse factors that may have potentially triggered these extinctions is debated, with the primary causes varying according to the continent under consideration and the scientific discipline involved in the research. Some argue that only human hunting mattered, others argue for the role of climate change, hyperdisease, habitat modification, or even extraterrestrial impact[1–10]. In the Americas, most extinctions occurred towards the end of the late Pleistocene, after the Last Glacial Maximum and near the time of the first widespread dispersal of humans from northeast Asia[11]. Because the loss of biodiversity in the Americas was severe and happened nearly synchronous with significant climatic changes and the initial influx of humans, the debate regarding the main factor(s) driving extinctions has been more controversial and persistent than for any other continent[3,9,12].

In recent years, it has become increasingly clear that extinctions were not homogeneous in the Americas[2,6,8], and the debate around this process was quite different in North America and South America. In North America, 70% (37 genera) of mammals with an average body mass over 44 kg (megafauna *sensu* Martin[13] or large mammals *sensu* Cione et al.[7]) disappeared mainly between 13 and 12 k cal BP[2]. This period corresponds with an abrupt cooling episode, the Younger Dryas, and the dispersal of Clovis culture, the earliest recognized widespread archeological techno-complex in the continent[11,14]. Clovis is characterized by a unique technology of fluted projectile points, strongly associated with the hunting of large mammals. In the span of a single millennium, Clovis spread rapidly over most of North America. Based on Clovis and Fishtail projectile point evidence, Paul Martin[13] formulated the challenging hypothesis of the "Pleistocene overkill", which postulated that the appearance of humans in the Americas was principally responsible for megafaunal collapse. At present, although this model has been strongly criticized[9], and the combined effect of climatic change and human action has been the most widely argued cause of extinctions, many scholars maintain that humans could have been the principal or necessary driver of the extinctions[3,15].

The situation is somewhat different in South America. There, late Pleistocene extinctions were more acute than in North America[1,6,7], with the loss of 82% (over 40 genera, *sensu* Cione et al.[7]) of megafaunal species, but less importance has been attributed by archeologist to humans as driving the process. The end of the Antarctic Cold Reversal, a South American post glacial cooling period earlier and less marked than the North American Younger Dryas[16], is synchronous with the beginning of the spread of the South American Fell or Fishtail projectile points (henceforth, referred to as FPP), which occurred during a time span that is similar to the Clovis period but followed a few centuries later[17–19]. FPP, usually but not always fluted, present a fainter and more irregular spatial distribution than Clovis in North America[17,20–22] and, although they are usually claimed to be linked with local megafauna, South American sites with remains of large mammals and clear evidence of direct exploitation by humans is elusive and restricted to a few species[6,8,23]. Based on such limited evidence, and assuming humans reached South America in "pre-Clovis" times[24], South American archeologists (1) maintain that FPP played an important but noncentral role during the colonization process—unlike Clovis in North America—, and (2) do not place humans squarely within the debate on the Pleistocene megafaunal extinction. Nevertheless, because climatic changes do not fully explain extinctions by themselves[12,25], a number of paleoecological and paleontological studies have increasingly credited humans as having a major role in the extinction process[1,7]. In this sense, it is remarkable that the megafaunal extinctions occurred just after the spread of FPP[24] and almost simultaneously with a significant slowdown of the human population growth rate. Although the new perspectives are seriously calling for the abandonment of the strict dichotomy between climate and humans as premier extinction drivers, the archeological and paleontological data have not been fully integrated using a quantitative approach. An updated study integrating both lines of evidence is necessary to rigorously evaluate the relationship between megafaunal and human dynamics, as well as to better define the actual role of humans in South America's late Pleistocene extinctions.

In this paper, we explore the temporal changes in the density of megafaunal species, including large and mega quaternary mammals, and FPP, as well as the variation in the potential distribution in geographical space. Although other projectile heads could have been used for hunting megafauna, FPP are the most abundant and widely distributed in early South America[17–19], and seem to have been a specialized weapon for that purpose[26]. So, we consider this artifact as a good empirical proxy for exploring the interaction between extinct large mammals and humans. On this basis, we evaluate with updated evidence and, from a continuous temporal and spatial perspective, the direct impact of human agency on the late Pleistocene extinctions in South America. If humans were the main drivers of the extinctions, we expect to see an inverse association between human and megafaunal population densities in time and a positive association in space.

Here, we investigate the temporal changes in density using Sum Probability Distribution of radiocarbon dates of archeological and paleontological samples (SCPD method[27,28]), whereas we explore the differences in potential for distribution in geographical space by employing Species Distribution Models (SDM)[29,30] and Stack Species Distribution Models (SSDM)[31]. For the last analyses, we explore the distribution of the FPP and ten species that present records of physical (stratigraphic) association with humans in the archeological record (*Hippidion saldiasi, Milodon darwini, Lama gracilis, Equus neogeus, Doedicurus clavicaudatus, Megatherium americanum, Glossotherium robustum, Notiomastodon platensis, Notiomastodon waringi,* and *Cuvieronius hyodon*) during the end of the late Pleistocene. Except for *G. robustum* and *D. clavicaudatus*, evidence of having been subject to human consumption or processing[6,23] is associated with all species. We compare megafaunal dynamics with the changes in human density and distribution using a screened database of archeological radiocarbon dates[24]. If megafauna were a central resource in human economy, we not only expect that humans impact megafaunal population dynamics; we also expect that changes in megafaunal density and distribution impact the human population. We also discuss changes in the archeological record in the framework of environmental and climate change inferred by previous work in South America[18,32,33]. We explore changes in density for the whole South America[34], but we focus our analyses on three relatively independent areas where evidence of coexistence between humans and megafauna is strongest: Pampa (including the Argentinean Pampas, South Brazil, and Uruguay), Southern Patagonia, and Andes (Supplementary Fig. 1). On this basis, we re-examine the main hypotheses regarding the late Pleistocene extinctions in South America and the importance of direct human impact on this process.

## Results

**Summed probabilities distributions of radiocarbon dates of megafauna, FPP, and all archeological sites.** The temporal changes in the density of megafauna and FPP were explored using the Sum Probability Distribution method (SCPD method)[27,28].

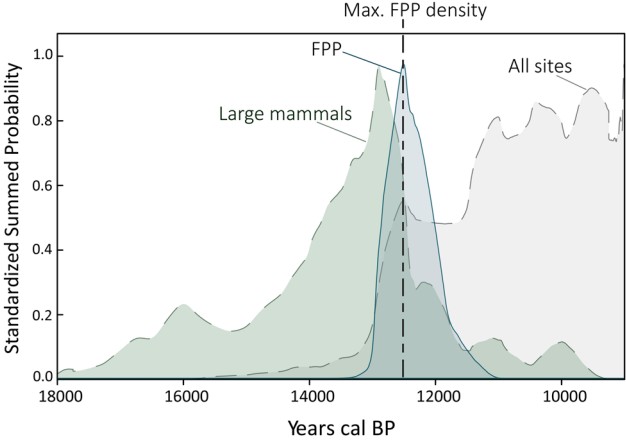

**Fig. 1 Summed probability distribution curves for South America.** The temporal change in the density of large mammals (or megafauna) (light green shading), FPP (light-blue shading), and archeological sites (beige shading) reflected in the SCPD curves for all of South America. *X* axis represents Calibrated years BP and *Y* axis the standardized summed probability.

These SCPDs were compared against the curve of human density in South America based on dates of archeological sites compiled by Prates et al.[24]. Our results (Fig. 1), show that the radiocarbon signal of large mammals around 18 k cal BP is extremely low in South America, but clearly increases from 17,5 k cal BP, and grows rapidly and steadily between 15,3 and 12,9 k cal BP. After 12,9 k cal BP, the SCPD curve shows a dramatic decline until 11,6 k cal BP. From this date onwards, only a few genera of extinct large mammals have been recorded[35] and most of the alleged early Holocene dates have recently been called into question[36] (Supplementary Fig. 2). Fishtail projectile point technology appeared in South America at ca. 13 k cal BP and shows a rapid amplification of density until reaching the distribution peak between 12,4 and 12,2 k cal BP (Fig. 1). From this time onward, a deep decline continues until the technology virtually disappears at ca. 10,9 k cal BP. Figure 1 also shows that the archeological—human—signal is low from the earliest appearance, ca. 15 k cal BP, to ca. 13 k cal BP, when it clearly increases. Around 12,5 k cal BP, the SPD curve for all archeological sites shows a peak followed by a slight decline that extends to 11,6 k cal BP before rising again (Fig. 1).

If we consider South America's main geographical regions separately (Supplementary Fig. 1 and Fig. 2), we observe significantly similar changes in density of megafauna (Fig. 2a) and FPP (Fig. 2b) when compared with the permutation test proposed by Crema et al.[28]. The density of megafaunal species displays differences among regions only in the earliest and latest dates, but a general agreement in the main density peaks is observed between ca. 13,5 and 12,5 k cal BP. The FPP's densities are almost identical among the regions (Fig. 2b). The changes in density for all archeological dates display some correlations with the megafaunal densities, with an apparent impact of the large mammal's density decline stronger in Patagonia than in the other regions, and a lower impact in Andes (Fig. 2c). Figure 2c also shows that the density of human signal after 11,5 k cal BP was significantly lower in Patagonia than in the other regions in relation to what can be observed in the previous millennium (12,5–11,5 k cal BP), with the highest values of density observed in the Andes.

**Spatial distribution of megafauna, FPP, and all archeological sites.** The potential spatial distribution of megafaunal species and

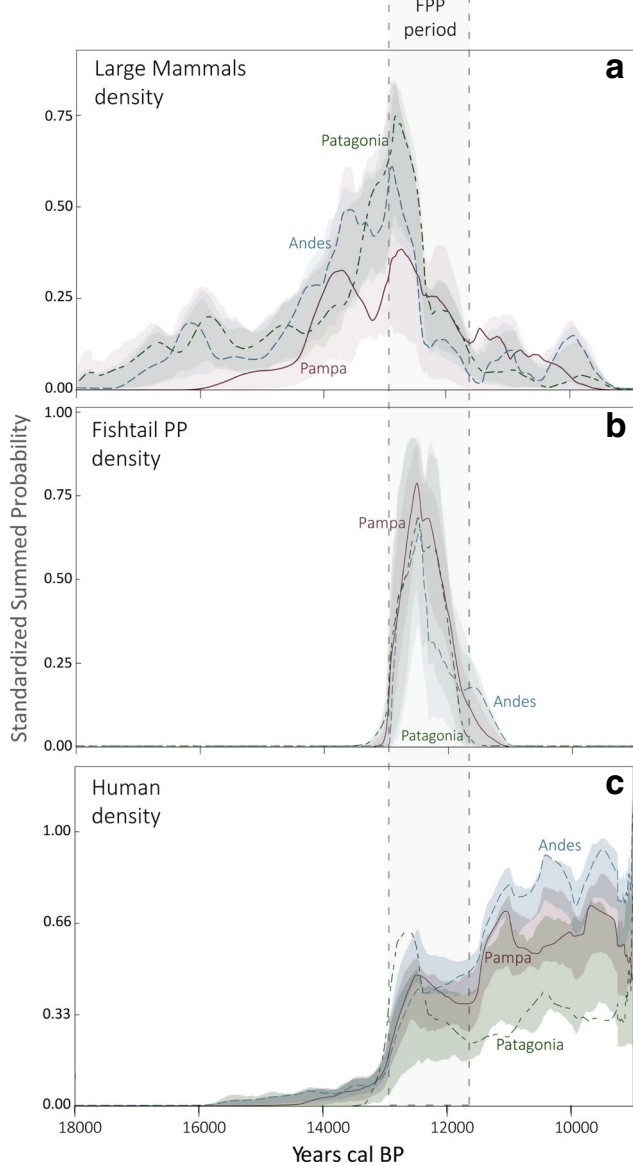

**Fig. 2 Regional differences in summed probability distribution curves within South America.** The temporal change in the density of large mammals (**a**), Fishtail projectile points (**b**), and all archeological sites (**c**) described using SCPD curves. Different regions are indicated in shading colors: Andes (light blues), Pampa (light red), Patagonia (light green). *X* axis represents Calibrated years BP and *Y* axis the standardized summed probability.

FPP was explored using a Species Distribution Modeling approach as implemented in MaxEnt[37], considering species and FFP occurrences between 18 k and 9 k cal BP, and the bioclimatic variables available in PaleoClim[38,39]. Figures 3–5 show the estimated distribution of the extinct megafaunal species with strong evidence of direct stratigraphic association with humans in archeological sites. The predictive capacity of the distribution models was high for all species, FPP, and archeological sites, with AUC values above 0.9 (*M. americanum*: 0.99; *G. robustum*: 0.98; *D. claviacaudatus*: 0.97; *E. neogeus*: 0.98; *L. gracilis*: 0.99; *M. darwinii*: 0.99; *H. saldiasi*: 0.99; *N. platensis*: 0.99; *N. waringi*: 0.97; *C. hyodon*: 0.91; FPP: 0.92; all archeological sites: 0.92). The modeled potential distribution maps for almost all megafaunal species display high values for Pampa and Patagonia. Based on

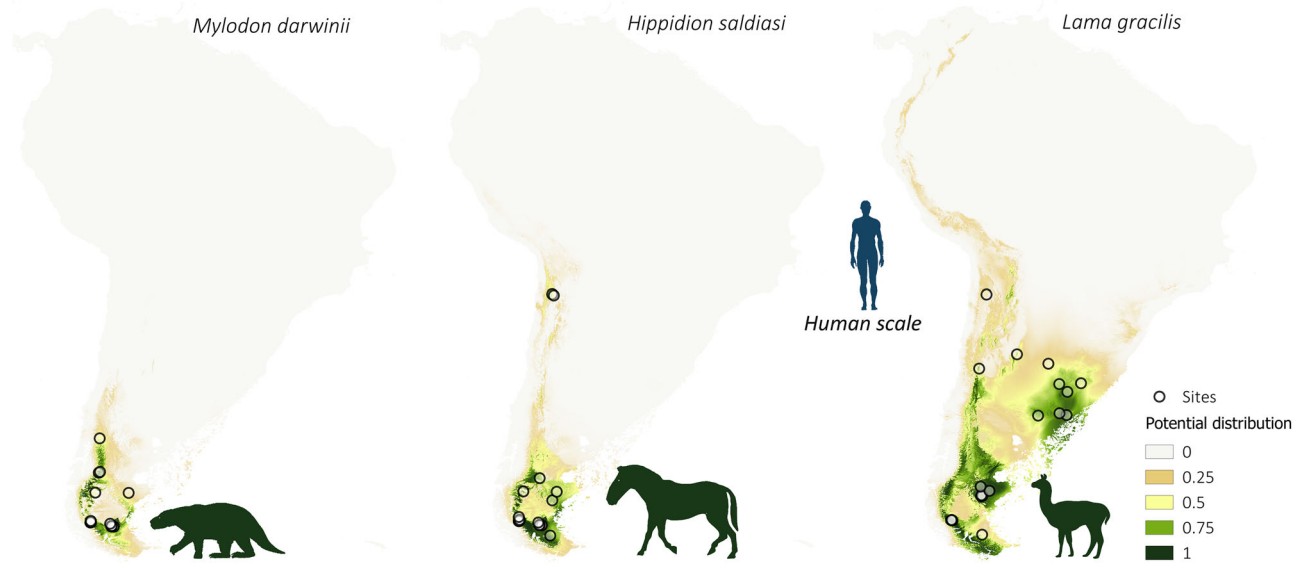

**Fig. 3 Spatial distribution of southernmost megafaunal species.** Potential distribution maps of the *Mylodon darwinii*, *Hippidion saldiasi,* and *Lama gracilis* during the late Pleistocene–early Holocene (18 k–9 k cal BP). Values of potential distribution vary from 0 (light beige) to 1 (dark green). Empty dots represent species occurrence sites. The taxonomy of *Lama gracilis* is under discussion. It has been mainly considered a vicariant of *Vicugna vicugna* in the low plains, but some morphological and molecular studies suggest that *L. gracilis* and *V. vicugna* may be the same species[65–67].

the values for the species, we can group them into three patterns: *M. darwini*, *H. saldiasi*, and *L. gracilis* with a predominant distribution in Patagonia and *L. gracilis* also in Pampa (Fig. 3); *E. neogeus*, *D. clavicaudatus*, *M. americanum*, and *G. robustum*, with a predominant distribution in Pampa (Fig. 4); and *N. waringi* and *C. hyodon* with a wider distribution than the other species, reaching most of Andes and northern South America, whereas *N. platensis* is distributed in Pampa and central Andes (Fig. 5).

Considering the modeled potential distributions of the megafauna, the local richness for all species together was estimated employing Stack Species Distribution Modeling[31] as implemented in SSDM. As Fig. 6a shows, higher values of species richness (>4) were estimated for Pampa, with intermediate values (2–4) in Southern Patagonia and Northeast Brazil, as well as lower values in Northern Patagonia, South-Central Andes, and North-Central Andes. The potential distribution of FPP also shows the highest values (between 1 and 0.6) in Pampa, with intermediate values in Patagonia, South, and Central Andes (Fig. 6b). The distribution of all archeological sites shows a similar pattern with FPP and species richness, but the values of potential distribution are lower in Pampa (Fig. 6c). The spatial density estimated from a kernel method in QGIS 3.14[40] displays a very similar pattern with relation to FPP and all archeological sites distribution (Supplementary Figs. 3 and 4). Particularly, in the 13–11 k cal BP period, the higher date density is observed for Pampa, Southern Patagonia, and South-Central Andes, where we describe the maximum values for species richness (Fig. 6 and Supplementary Fig. 4).

**Niche overlapping analysis: megafauna vs. FPP and all archeological sites.** We estimated niche overlapping using the *I* similarity index[41], which varies between 0 (no overlap) and 1 (identical distributions). The similarity index of all archeological sites and FPP with megafaunal species (Supplementary Table 1), as well as among them, is plotted in Fig. 7. Moreover, a non-metric multidimensional scaling analysis (nm-MDS) was performed to summarize the similarity matrix. The boxplot of *I* similarity index (Fig. 7a) shows on average larger similarity values

between FPP and each large mammal species than observed among all species. *M. americanum*, *G. robustum*, *D. clavicaudatus*, *N. platensis*, *L. gracilis,* and *E. neogeus*, with high values of potential distribution in Pampa, display the highest/similarity values with the FPP distribution (Supplementary Table 1). The nm-MDS plot shows that the FPP distribution is intermediate among all species, being closer to the species with high values of potential distribution in the Pampas (Fig. 7b). The nm-MDS scores also show a clear separation between three clusters of species grouped according to their main distribution areas: Patagonia (*M. darwinii* and *H. saldiasi*), Pampa (*M. americanum*, *G. robustum, D. clavicaudatus,* and *N. platensis*), and northern South America (*N. waringi* and *C. hyodon*; Figs. 6 and 7b). *E. neogeus* and *L. gracilis* are close to the Pampa and Patagonia, respectively. All archeological sites are closer to the FPP position, but they are closer overall to the *N. waringi* and *C. hyodon* species distributions (Figs. 6 and 7b).

**Discussion**

The SCPD curves of the density of megafauna through time (Figs. 1, 2 and Supplementary Fig. 2) reflect a significant increase of the fossil record all over South America shortly after the Late Glacial Maximum (ca. 17,5 k cal BP). Although this pattern is probably linked to the expansion of large herbivorous mammals and their predators during the end of the late Pleistocene[42], it could be also influenced by taphonomic biases or the loss of fossil specimens over time. If the expansion was real, it could be related to the favorable environmental condition for herbivores after the end of the Last Glacial Maximum[42,43], and especially in the Pampas and Patagonia regions. According to our results, this expansion continued ~4,5 k years until a sudden and deep decline of the radiocarbon dated fossil record at ca. 12,9 k cal BP. Interestingly, when we compare the FPP and megafaunal SCPD curves, we observe that the sudden decline of the megafauna's density started right after the appearance of FPP technology in South America, ca. 13 k cal BP. Moreover, as shown by Figs. 1 and 2, from that moment an explosive increase of the FPP occurs closely followed by a steep decline of the megafauna. The frequency of FPP increases for ~600 years (until ca. 12,4 k cal BP)

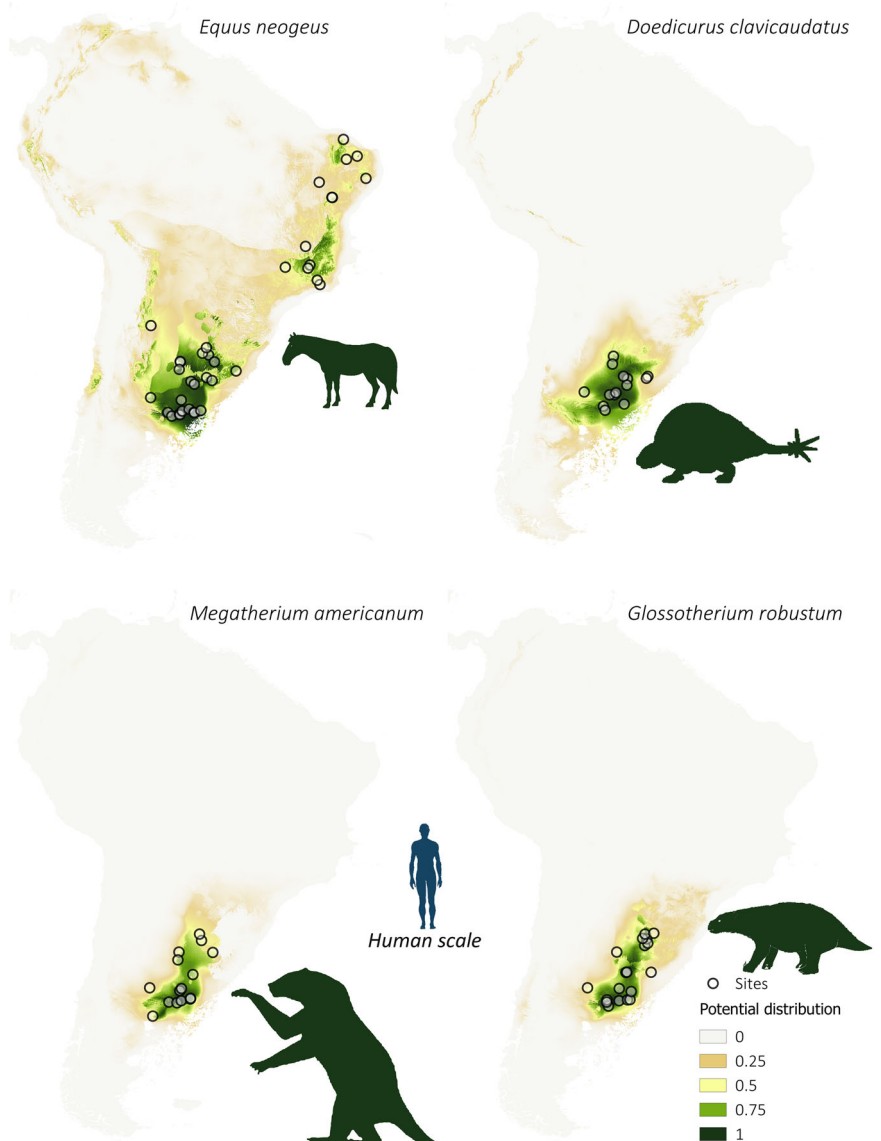

**Fig. 4 Spatial distribution of Pampean megafaunal species.** Potential distribution maps of *Equus neogeus*, *Doedicurus clavicaudatus*, *Megatherium americanum*, and *Glossotherium robustum* during the late Pleistocene–early Holocene (18 k–9 k cal BP). Values of potential distribution vary from 0 (light beige) to 1 (dark green). Empty dots represent species occurrence sites.

before declining rapidly, similar to what is observed for the megafaunal curve. Fishtail projectile points and megafauna virtually disappear together from South America ~10,9 k cal BP, supporting the hypothesis that FPP technology was directly linked to megafauna extinction.

The changes in the density of FPP over time not only seem to be related to the density of megafauna, but also to important demographic changes in South American human populations[24]. As shown in Fig. 1, during the initial peopling of the continent, between 15 and 13 k cal BP, the archeological signal (and probably human population density[24]) stayed extremely low, until an irruptive growth dynamic occurred just after the FPP spread over southern South America. The tight fit between the behavior of both late Pleistocene phenomena, FPP and human population expansion (Fig. 1), may indicate that rapid and successful dispersal of FPP technology drove the high rate of population growth of the earliest hunter-gatherers. Or, from a more conservative perspective, that users of FPP technology were the first colonizers of South America[14]. Either way, the explosive population growth of the human population stopped suddenly at ca.

12,5 k cal BP, just before FPP reached the peak of the distribution curve and started to decrease, a few centuries after the initial decline of megafauna. These temporal patterns resemble what is observed in North America for Clovis technology[15] although FPP technology is somewhat younger and overlaps with the end of Clovis, as recently observed by Waters et al.[19]. However, whether FPP technology is a southern cultural expression of Clovis or a result of independent innovation is an open question. What is clear in our temporal density results is that human demography in South America during the end of the late Pleistocene was related to the changes in the density of several megafaunal species and to the expansion of FPP technology.

The potential distribution models generated for the different species of megafauna and for the FPP (Figs. 3–6) show large spatial overlapping in South America, except for some gomphotheres (*C. hydon* and *N. waringi*) whose distribution does not coincide with FPP. This may suggest these species were not the principal prey of FPP hunters. The largest values of potential overlap are in the Pampas (including Argentina, Uruguay, and Southeast Brazil) and Southern Patagonia, but with smaller values

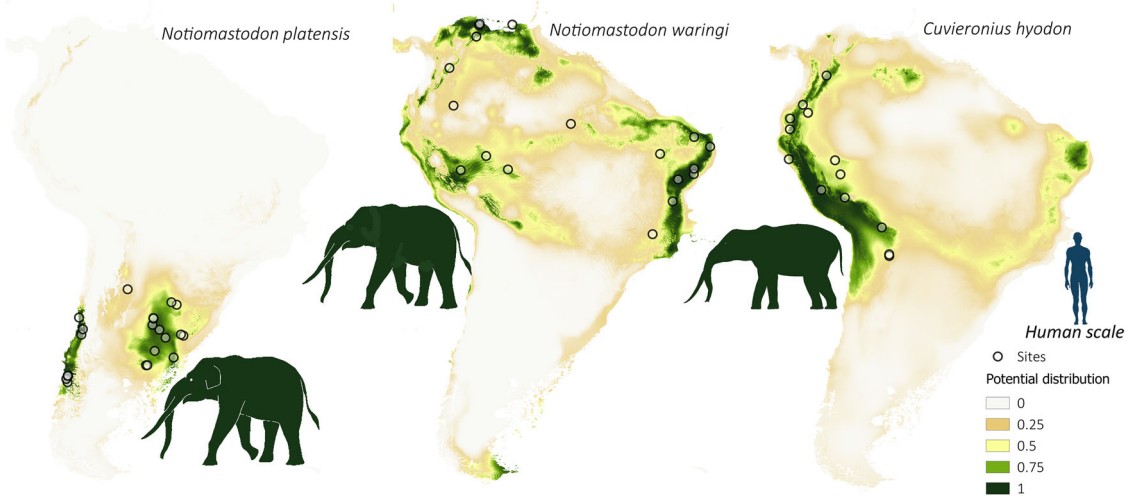

**Fig. 5 Spatial distribution of northernmost megafaunal species.** Potential distribution maps of *Notiomastodon platensis*, *Notiomastodon waringi*, and *Cuvieronius hyodon* during the late Pleistocene–early Holocene (18 k–9 k cal BP). Values of potential distribution vary from 0 (light beige) to 1 (dark green). Empty dots represent species occurrence sites. Mothé et al.[68] has proposed that *N. platensis* and *N. waringi* are the same species, but we follow Prado and Alberdi[69] who suggest they represent two different species or two geographic variants.

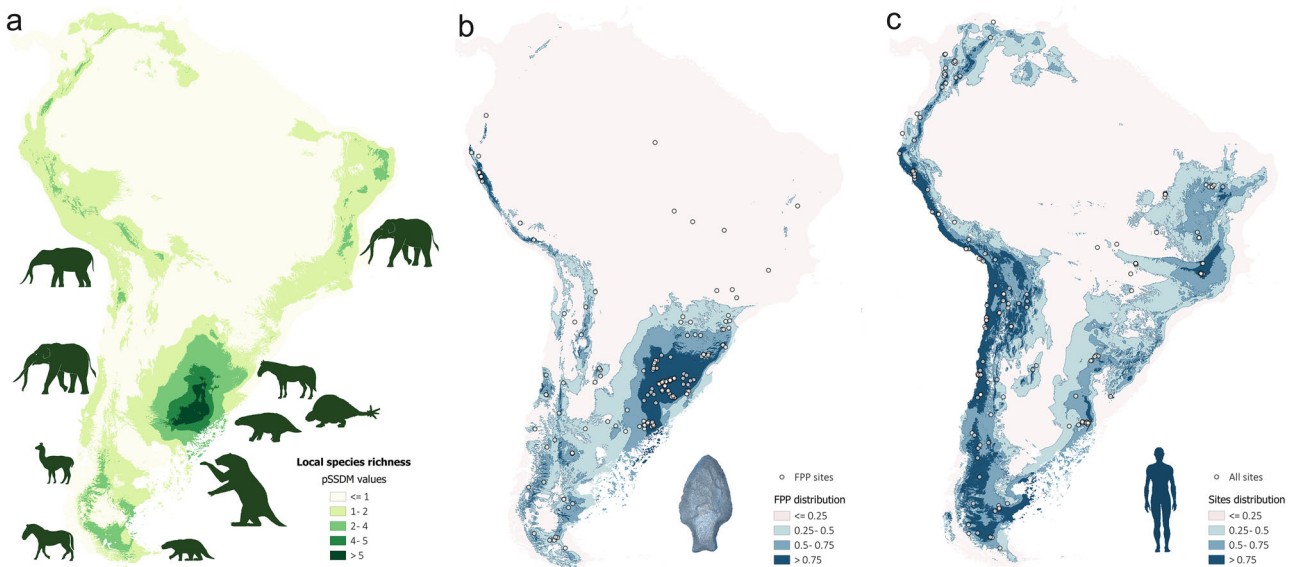

**Fig. 6 Spatial distribution of species richness, FPP, and human occupations.** Maps describing the local species richness of extinct large mammals (18 k–9 k cal BP) (**a**), the potential for distribution of Fishtail projectile points (13 k–10,9 k cal BP) (**b**), and all archeological sites (13 k–11 k cal BP) (**c**). Local species richness varies from 1 (light green) to >5 (dark green), and the potential distribution of both FPP and all archeological sites vary from 0 (light blue) to 1 (dark blue).

in the latter. The Andes appear as a marginal area for both FPP and the megafauna species analyzed here. This correspondence is clearly observed in Fig. 6, which shows that the largest potential distribution values for FPP (Fig. 6b and Supplementary Fig. 3) coincides with the areas with the largest local richness of megafauna species (Fig. 6a), as well as the areas where high density of human occupation is observed (Fig. 6c and Supplementary Fig. 4). However, the human populations are distributed more widely than FPP technology in South America. Although distributional overlap is only an indirect estimation of trophic interaction[44], this measurement strongly supports the importance of FPP technology for hunting species of large mammals during the late Pleistocene.

The areas with higher values of potential for FPP distribution and species richness are characterized by the predominance of open environments: grassland steppes in the Pampas and grassland cold steppes in Patagonia, to which the large mammal species analyzed here were better adapted and which they occupied with greater concentration at the end of the Pleistocene[7,42,43]. Moreover, the results of the niche overlap analysis show that the distribution of FPP is focused in a central geographic sector with respect to the spatial distributions of the large mammal taxa. They show greater proximity to the Pampean species as well as to Patagonian ones. This implies that the FPP, in addition to expanding rapidly and with a distribution similar to that observed for megafauna, did so mainly in the open environments. This reinforces the hypothesis that FPP were designed and used for hunting megafauna.

On this basis, we suggest that: (1) if South America were colonized ca. 15 k cal BP, when the abundance of megafauna was relatively high, their populations were unaffected because humans must have been generalist hunter-gatherers; (2) when megafauna was at

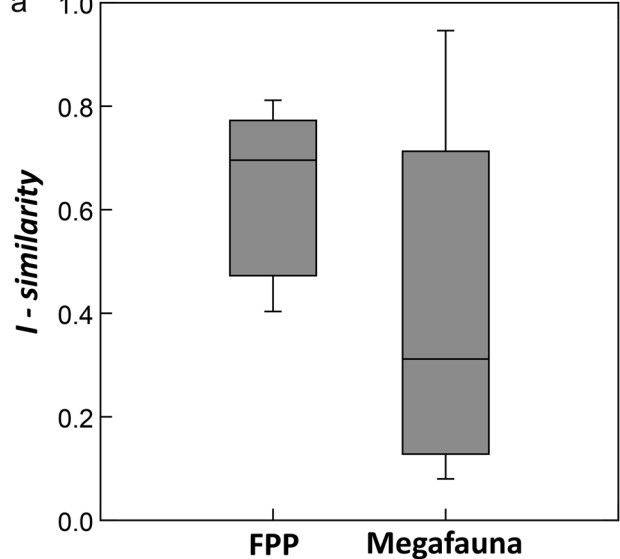

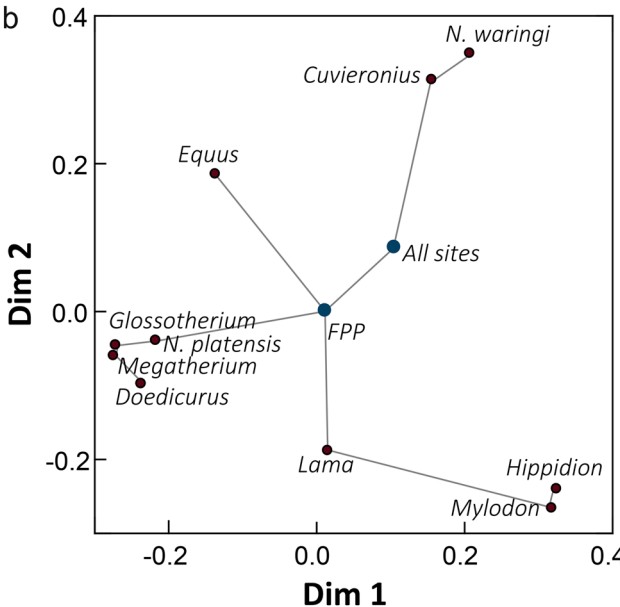

**Fig. 7 Similarity in the spatial distribution of FPP, humans, and megafaunal species.** Boxplot (**a**) of the values for $I$ similarity index describing niche overlap between FPP and extinct large mammals ($n = 10$), as well as all extinct large mammal species ($n = 45$). This graph shows the minimum and maximum values (whiskers), the first and third quartiles (bounds of box), and the median (centerline). nm-MDS ordination (**b**) describing the similarity in niche among all archeological sites, FPP, and all extinct large mammal species. The minimum spanning tree describing the $I$ similarity matrix is shown.

gomphotheres, also disappeared after megafaunal extinction. It is remarkable that when megafauna and FPP disappear, an abrupt deceleration and subsequent decline occurred in the growth of the human population throughout the continent. Although this process is observed in all regions of South America, in Patagonia it seems to have been more drastic, with a stronger and longer decline over time (Fig. 2). This process could be related to the occurrence of a population overshoot, in which humans exceeded carrying capacity in the late Pleistocene ecological community. This could have generated a population collapse of the megafaunal prey, with the consequent drop in the carrying capacity of the environment for big-game hunters and the subsequent demographic decline in the populations of the main predator species in the trophic network, the humans.

Several important changes in climatic and environmental conditions occurred at about the same time as the extinction of large mammals and the widespread dispersal of hunter-gatherers carrying FPP technology in South America. An increase in temperatures began after the Last Glacial Maximum, ca. 18–17 k cal BP[16,32,47], and ended with the Antarctic Cold Reversal (ACR; Fig. 8; ca. 14,7–13 k cal BP)[16,47]. This post glacial cooling period occurred when the megafauna reached their maximum growth in density (Fig. 8) and coincided with the North American Bølling–Allerød warm stage[48]. After the ACR, a warming period began at ca. 13 k BP, which coincided with the Younger Dryas[16] cooling period in North America. Terrestrial paleovegetation proxies do not suggest an abrupt environmental change in southern South America at the end of the ACR, but cold/cool and wet conditions persisted until 11,8 BP[49,50]. Open grasslands, the preferred habitat of the megafauna[7,42,43], seem to have retracted and reached their current distribution after 12,4–11,5 k cal BP in different areas of Patagonia[7,32,33,42,51], and probably later in the Pampas[52]. In addition, Pampa and Patagonia suffered a significant reduction of territory due to the rise in sea level[42,43].

All these climate and environmental changes could have played the central role in Pleistocene extinctions together with humans, but there are several aspects to consider. First, the gradual effects of the end of the ACR climatic event in Pampa and Patagonia (Fig. 8)[16,32,49,50] do not seem congruent with a sudden decline of megafauna ca. 12,9 k cal BP (Fig. 8). Second, the shrinking of the grasslands steppes—the most favorable environmental conditions for large mammals—from southern South America occurred between 12,4 and 11,5 k cal BP[32,33,51], 1400–500 years after the megafaunal decline (12,9 k cal BP) in South Patagonia; in the Pampas the change seems to have occurred even later. Finally, whereas in North America the initial decline of the megafauna at 13 k cal BP is contemporaneous with the beginning of the Younger Dryas post glacial cooling period at 13–11,7 k cal BP, in South America the initial decline of megafauna (12,9 k cal BP) is contemporaneous with the beginning of the warming period starting after the ACR, at ca. 13 k cal BP. Although this change could have affected the megafauna by exerting under high ecological stress, the animals did not become extinct until humans using Fishtail projectile points appeared, suggesting again that human agency could have been a determining factor driving their extinction.

Our results support the hypothesis that early hunter-gatherers were an important driver of the megafauna's extinctions in South America. Nevertheless, there are some still unanswered questions: (1) if human arrived in South America ca. 15.5 cal BP[24], why did they not have an impact on the megafaunal expansion until 12,9 k cal BP, despite having coexisted with them for ~2500 years; (2) why is direct archeological evidence of megafaunal exploitation so rare[6,9,23,53]; and (3) why would so many megafaunal species become extinct during the end of the Pleistocene when they do not seem to have been exploited by humans, and why

maximum density in the open grassland steppe environments, mainly Pampa and Patagonia, the hunter-gatherers started to prey on them using FPP; and (3) a little later, ca. 12,9 k BP, the growth trend in number and density of megafaunal species stopped abruptly and began to decline. This drop in the density of megafauna had effects a few centuries later, ca. 12,4 k BP, in the adaptive pattern of the hunter-gatherers who used the FPP technology, which diminished rapidly until it completely disappeared when the megafauna became almost extinct. Other types of projectile heads (e.g., projectile points made of bone and El Jobo projectile points)[45,46], which were probably used at the same time in preying on some species of

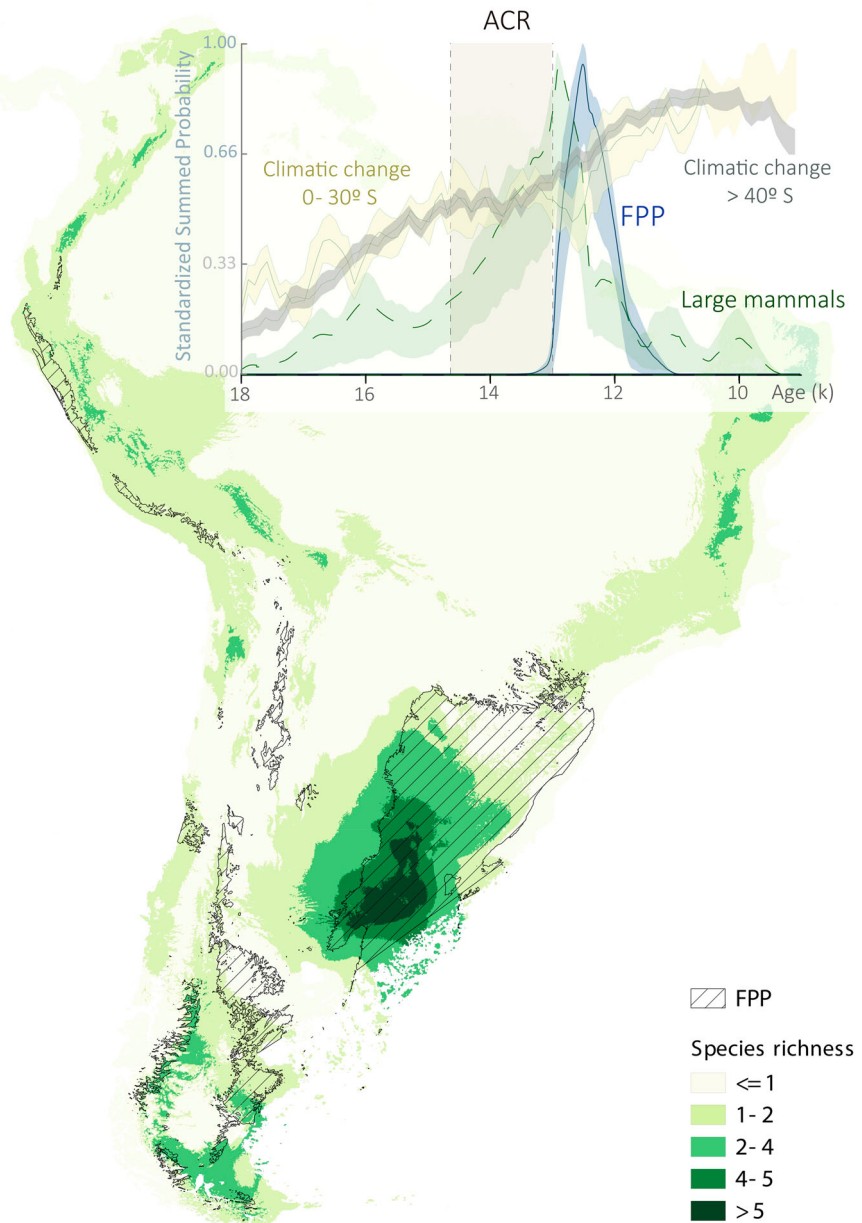

**Fig. 8 Summary of spatial (map) and temporal (inset graph) results.** Local species richness map of extinct large mammals and the calculated potential distribution of FPP, together with the temporal change in the density of both variables for all of South America and the climatic changes following Pedro et al.[16]. Striped areas in the map represent the potential distribution of FPP. Local species richness of extinct large mammals varies on the map from 1 (light green) to >5 (dark green). Relative climatic changes (temporal temperature curves) are indicated in beige (0–30° S) and gray (>40° S), and the ACR period in dark beige. The temporal change in the density of large mammals is represented by light green shading area (lower and upper bounds of the simulated envelope) and dashed green line (observed summed probability). The temporal change in the density of FPP is represented by light-blue shading area (lower and upper bounds of the simulated envelope) and blue line (observed summed probability). In the inset graph X axis represents Calibrated years BP and Y axis the standardized summed probability.

guanaco, probably the most heavily preyed-upon species, did not suffer extinction. Regarding the first questions, our analysis suggests that humans rapidly and strongly affected the mega-faunal population, not from the time of the first arrival, but after the FPP big-game hunters came onto the scene. A simpler explanation for why archeological sites are so rare and why megafaunal populations are unaffected until 12,9 k cal BP could be that pre-Clovis in South America is not real. But it seems more reasonable to us that the food preferences of the earliest-dispersing humans have involved a broad spectrum foraging behavior, and their initial population density may have been too

low to affect the large mammals. Regarding the second and third questions, recognizing that humans played a principal role in the extinction process requires neither high archeological visibility of hunting nor massive predation on all the extinct species. Large mammal populations were already adaptively vulnerable because of climatic and environmental changes, partly because most of these large-bodied species had low reproductive rates[7], factors that were compounded by deleterious anthropogenic effects on the environments[1,54]. Relatively moderate or even low levels of human predation on a few species could have strongly impacted trophic networks. Thus, the environmental imbalance could have

contributed to the extinctions of hunted species and others impacted by the compounded changes. As Pires et al.[55,56] have shown, the arrival of human and their predation of a few species of large mammals in Southern Patagonia, or even on a single one (e.g. *Lama guanicoe*) would have created multiple indirect effects on species, such as *M. darwini* and *L. gracilis*, probably driving them to extinction. Specialized hunting of only a few species, employing FPP technology, could have set the baseline for the massive collapse of the megafaunal community and almost all large mammals. This process, together with the low likelihood that large prey would have been transported by early humans to remote camps[6,57], and the extremely low probability of finding evidence of predation by humans on megafaunal species due to sampling/taphonomic issues, including diffusion due to the processing of carcasses[4,58], could partially explain the apparent contradiction between our results and the lack of larger and stronger archeological evidence of megafaunal exploitation in South America. Although we are unable to explain why the guanaco has not gone extinct, this species was more geographically dispersed during the final Pleistocene[59] than most of the other large mammal species. This probably made the animal a better survivor than others. Furthermore, some have suggested that the endemic Patagonian guanaco did indeed become extinct but the region was re-populated by a subspecies from the Central Andes[60], similar to what might have happened if *Lama gracilis* is really a subspecies of *Vicugna vicugna*.

In conclusion, since Martin's proposal[13] regarding the rapid spread of big-game hunters in the Americas armed with fluted points (Clovis in the Northern Hemisphere and Fishtail in the South), the central role of humans as agent of megafaunal extinction had been little considered by South American archeology. The strong evidence of pre-FPP occupation and the weak direct evidence of interaction between humans and extinct large mammals in the archeological record has led to a consensus among archeologists that humans played a secondary role or none at all in the process. Here, however, we have shown that the available archeological and paleontological evidence is compatible with a significant impact of human hunting on megafaunal depopulation, not necessarily an overkilling effect. Particularly, we have demonstrated there is a strong spatial and temporal relationship between FPP technology, which appears to be directly related to large mammal hunting, with the density and distribution of large mammal species, as well as with the distribution and fluctuations in human demography during the critical period. Moreover, the decline in megafaunal density seems not to fit well with the timing of significant climatic and environmental changes in southern South America.

On this basis, we propose (1) that human predatory behavior was the main factor driving the megafaunal decline in South America, and (2) that the late Pleistocene environmental changes and the indirect effect of humans on the ecological web probably generated the conditions resulting in the massive collapse of the megafaunal community. Future studies are necessary to explore in-depth the individual demographic trajectories of all extinct megafaunal species and their relationships with climatic, technological and human demographic changes, as well as the relative importance of direct and indirect effects of human hunting.

## Methods

The temporal changes in density and the spatial variation in the potential distribution of megafaunal species and FPP during the late Pleistocene and early Holocene were explored employing radiocarbon dates compiled by Prates et al.[24]. These radiocarbon dates were complemented with dates for paleontological megafauna recently published[51,60–62] (Supplementary Data 1 and 2). The radiocarbon dataset was built in agreement with standard validation criteria used in archeology, such as it was described in Prates et al.[24]. In addition, we compiled a dataset of megafauna and FPP without absolute dates, but with relative date (end of

late Pleistocene) and precise geographic location using as a reference several recent works[8,20–22,60–63] and Paleobiology Database (https://paleobiodb.org/), but checking and expanding the data in the original publications (Supplementary Data 3 and 4).

The temporal changes in the density of FPP and megafauna were explored by using the SCPD method[27,28]. The SCPD was reconstructed using calibrated dates binned by site in intervals of 200 years, as well as 500 years for the window size of the moving average for smoothing. The FPP and megafauna SCPDs were compared against the curve of human density in South America based in the complete dataset for archeological sites compiled by Prates et al. [24]. Moreover, the differences in SCPD among the three geographical regions studied in human density, FPP and megafauna were explored with the permutation test proposed by Crema et al.[28], using 1000 permutations. The radiocarbon dates were calibrated using the Southern Hemisphere SHCal 20 curve and all SPD estimated with the package *rcarbon* for the R software 4.0[64]. Because almost all megafaunal records occur between 18 and 9 k cal BP, we restricted the plots to this period.

The spatial variation in the FPP and megafauna was explored by using an SDM approach[30]. These models are widely used in paleoecology[60,63]. For comparative purposes we also explored the potential distribution for human archeological sites dated between 13 and 11 k cal BP, as well as the spatial kernel density of dates using QGIS 3.14[40]. We used a maximum entropy approach to species and projectile point distribution modeling based on presence-only data, as implemented in MaxEnt[29,37]. We consider as predictor the bioclimatic variables available in PaleoClim for the 14,7–12,9 k cal BP interval (http://www.paleoclim.org/)[38,39]. The performance of each model was evaluated using the AUC statistic (i.e., area under curve)[30], which measures the explicative power of the model varying between 0 and 1 (the better predictive value). We also explored the combined distribution of megafaunal species to produce a community-level model by using an SSDM approach[44]. The species distribution and stacked species distribution models were generated using the MaxEnt algorithm implemented in the packages *dismo* and *SSDM* for the R software 4.0[44,64]. The local map of species richness was estimated using the method of summing continuous habitat suitability maps (pSSDM)[44].

Finally, we estimated the niche overlapping using the *I* similarity index proposed by Warren et al.[41]. This statistic is based in the Hellinger distance that quantifies the similarity between two probability distributions, being related to the Euclidean distance for discrete distributions. The *I* similarity index is a modification of this distance to compare Hellinger-based results to more conventional ecological measures of niche overlap, varying between 0 (no overlap) and 1 (identical distributions). The values of similarity indexes for potential species and FPP distributions were plotted and a no-metric Multidimensional Scaling (nm-MDS) was performed to summarize the similarity matrix. The *I* niche overlap similarity index was estimated using the package *dismo* for the R software 4.0[44,64]. The nm-MDS analysis was performed in the software PAST 4.0.

**Reporting summary**. Further information on research design is available in the Nature Research Reporting Summary linked to this article.

## Data availability

All relevant data are within the paper and its Supplementary Data files.

## Code availability

R scripts to analyze the radiocarbon and occurrence species data are available as Supplementary Code 1.

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

## Acknowledgements

We thank Gustavo Politis, Valeria Bernal, Gustavo Martínez, Diego Rindel, Matías Medina, Laura Miotti, Gary Haynes, and Alfred Rosenberger for their helpful suggestions and comments on preliminary versions of the manuscript. This work was supported by Consejo Nacional de Investigaciones Científicas y Técnicas (CONICET), Universidad Nacional de La Plata (UNLP), and Agencia Nacional de Promoción Científica y Tecnológica (ANPCyT) through projects PIP-244/2015, PIP-729/2015, PI-11-N932/2020, and PICT-3645/2015.

## Author contributions

L.P. contributed to conceptualization, writing, and editing. S.I.P. contributed to conceptualization, formal analysis, and writing.

## Competing interests

The authors declare no competing interests.
