## [Peer Review File · Nature Communications]

Reviewers' Comments:

Reviewer #1:

Remarks to the Author:

The take-home message I have from this submission is that populations of the largest mammals grew quickly after the LGM then began to decline at about the time FPP human populations grew rapidly -- before the climate-caused decline in open habitats. The human populations using FPP technology then later declined. A logical implication is that FPP humans targeted the declining megamammals.

Recommended changes:

Line 34 -- references 1-4 seem a better fit for this statement. Also, add Wolf and Broughton's 2020 paper (JAS 119) about the associational critique.

Line 51-52: the statement about "first big-game hunters in the world" is embarrassing wrong -- it ignores the abundant and much older record of mammoth-hunting (and horse-hunting, bison-hunting, and reindeer-hunting) in Eurasia.

Line 73: replace "them self" with themselves (one word).

Lines 84-86: the sentence fits better in a Materials and Methods section, which should be moved to precede the Results section.

Lines 116-120: this sentence should be in the Results section of the paper.

Lines 122-126: This should be placed in a Materials and Methods section.

Lines 166-170: This should be placed in a Materials and Methods section preceding the Results.

Lines 200-203: This should be placed in a Materials and Methods section preceding the Results.

Fig. 5: the 'human scale' image makes the proboscidean silhouettes look relatively small. Were the fossil species actually that small?

Lines 222-224: This should be placed in the Materials and Methods section preceding the Results.

Line 226: after "no-metric" insert (nm) (in parentheses).

Line 229: a word is missing between "Particularly," and "M. americanum".

Line 239: the antecedent is unclear for the phrase "it is closer..."

Line 250: Does the date '17,5 k years BP' refer to radiocarbon years?

Lines 425+: A Materials and Methods section usually precedes the analytical results and conclusions. Is this the journal's format?

reviewed by G. Haynes

Reviewer #2:

Remarks to the Author:

I consider that the study you have made is novel and contributes to the debate around the LQE in South America. Even if I have been working in this field for quite a few years now, still surprises me

how the debate around the causes of megafaunal extinction remains open and heated. In this context, every good contribution (such as this present one) is one more step to reach the definite answer (if possible).

I have several comments, suggestions and concerns about your manuscript that are detailed in the attached annotated PDF. I hope they are helpful and contribute to make your work better.

Best regards,

Natalia A. Villavicencio

Reviewer #3:

Remarks to the Author:

Based on the objections mentioned above, it is up to the Nature editors to accept this paper for publication.

In the following revision the numbers indicated the lines.

20: I do not know if the in the abstract acronyms may be used. However, please explain before what "FPP" means.

44: says: ".13 and 12 k years BP..". Please, specify is they are calibrated years.

In the supplementary table regarding the list of sites with fishtail points, to the data recorded from Uruguay, in many of them, the authors refer Weitzel et al. (2018) paper as the main reference. However, the primary source (Nami 2013) of those findings is not properly provided. I suggest citing the primary source from that data deserves to be cited. Actually, in that publication, tables are providing the precise origin, metric data, and other observations of each specimen. Also, the primary source to access the detailed data of most detailed published Uruguayan fishtail points are given in several papers published by Nami. All of them provide very much detail for the Uruguayan Fishtails points.

A similar situation occurs in the supplementary table of radiocarbon dates. Indeed, when it provides the dates from Cueva del Medio depicted in rows 22 to 28 whose laboratory identification is NUTA and Gr-N, the authors provides as main reference the Martin et al. (2015) and does not cite the primary source of the dating, which is Nami and Nakamura (1995). Curiously, this original article is only referred to in one of the Cueva del Medio dates obtained in the same Japanese laboratory. The dates processed in the Groningen lab are provided in Nami and Nakamura (1995), and if the authors wish a more recent reference to use, the totality of the dates for the Fishtail occupation are available in Nami (2019). They are also referred to in other articles (Nami 2007, and 2017).

Nami, H. G. 2007. Research in the Middle Negro River Basin (Uruguay) and the Paleoindian Occupation of the Southern Cone. *Current Anthropology*, 48, 164-176. <https://doi.org/10.1086/510465>

Nami, H. G. 2013. Archaeology, Paleoindian Research and Lithic Technology in the Middle Negro River, Central Uruguay. *Arch. Disc.*, 1, 1-22.

Nami, H. G. 2017. Silcrete as a valuable resource for stone tool manufacture and its use by Paleo-American hunter-gatherers in southeastern South America. *Journal of Archaeological Science: Reports*, 15, 539-560. <http://dx.doi.org/10.1016/j.jasrep.2016.05.003>

Nami, H. G. 2019. Paleoamerican Occupation, Stone Tools from the Cueva del Medio, and Considerations for the Late Pleistocene Archaeology in Southern South America. *Quaternary*, 2, 28.

Special Issue "Changing Nature of Socio-Ecological Interactions in the Americas: From PalaeoAmerican through to Present Day". doi:10.3390/quat2030028

Nami, H.G., Nakamura, T. Cronologia radiocarbonica con AMS sobre muestras de huesos procedentes del sitio Cueva del Medio (Ultima Esperanza, Chile). An. Inst. Pat. Ser. Cs. Soc. 1995, 23, 125-133.

Reviewer #4:

Remarks to the Author:

This paper should be published with minimal revisions. The authors show a clear correlation between increasing human and declining megafaunal populations in the Late Pleistocene of South America. This is an important result and one that deserves publication. Specific comments follow:

- Lines 32-33 – the authors state that "there is a general agreement that the combined role of human activity – particularly hunting – and climate change related to late glacial fluctuations were the major drivers of extinctions on a global scale." I am skeptical of this. Some people would argue that only human hunting mattered. Others would argue for the role of hyperdisease, habitat modification, or even extraterrestrial impact. I think it would be better to state that cause continues to be debated.

- Lines 67-69 "Based on this lack of evidence, along with the general agreement regarding humans reaching South America in "pre-Clovis" times". Again, all archaeologists do not agree with this sentiment. In fact, if one considers the possibility that South America had no human population until the appearance of fishtail points, the patterns identified in this paper make a lot more sense. As an example of a recent Clovis-first study, see for example: Fiedel, S.J. (2017) The Anzick genome proves Clovis is first, after all. *Quaternary International* 444, 4–9. Also, the Surovell et al. paper cited (21) makes a similar argument.

- Lines 92-95, I recommend stating this sentence differently. This sentence forms the basis of hypothesis testing. It could be better stated as, "If humans caused the extinctions, we expect to see an inverse correlation between human and megafaunal populations in time and a positive correlation in space."

- Lines 108-109 – "If megafauna were a central resource in human economy, we expect that changes in megafaunal density and distribution must have impacted human population dynamics." I would expect the opposite, wouldn't human populations impact megafaunal population dynamics?

- Lines 126-130. Here it suggested based on an SPD that megafauna populations increase through time from ca. 18,000 to 13,000 BP. This pattern, however, has been repeatedly observed in other contexts and is almost certainly caused by taphonomic bias, or the loss of fossil specimens over time. See the following:

- Bluhm, Lara E., and Todd A. Surovell. "Validation of a Global Model of Taphonomic Bias Using Geologic Radiocarbon Ages." *Quaternary Research*, 2018.

- Surovell, T. A., and Spencer R. Pelton. "Spatio-Temporal Variation in the Preservation of Ancient Faunal Remains." *Biology Letters* 12 (2016): 20150823.

- Surovell, Todd A., and P. Jeffrey Brantingham. "A Note on the Use of Temporal Frequency Distributions in Studies of Prehistoric Demography." *Journal of Archaeological Science* 34 (2007): 1868–77.

- Surovell, Todd A., Judson Byrd Finely, Geoffrey M. Smith, P. Jeffrey Brantingham, and Robert L. Kelly. "Correcting Temporal Frequency Distributions for Taphonomic Bias." *Journal of Archaeological Science* 36 (2009): 1715–24.

- Surovell and Pelton (2016) have specifically developed a taphonomic correction curve for South American Pleistocene fauna. I would not insist that the author use it, but they should at least acknowledge that the apparent growth in the number of dated specimens might have nothing to do with population dynamics but is likely instead just preservation or taphonomic bias.

- The same issue should be acknowledged again in lines 248-252.

- Lines 268-271 - "As shown in Fig. 1, during the initial period of peopling, between 15 and 13 k years BP, the archaeological signal (and probably human population density stayed extremely low, until an irruptive growth dynamic occurred just after the spread over southern South America." I would

suggest that the authors at least consider the possibility that preClovis in South America is not real. This would provide a much simpler explanation for why archaeological sites are so rare and why megafaunal populations are unaffected.

RESPONSE TO R1'S COMMENTS

The take-home message I have from this submission is that populations of the largest mammals grew quickly after the LGM then began to decline at about the time FPP human populations grew rapidly -- before the climate-caused decline in open habitats. The human populations using FPP technology then later declined. A logical implication is that FPP humans targeted the declining megamammals.

1) Line 34 -- references 1-4 seem a better fit for this statement. Also, add Wolf and Broughton's 2020 paper (JAS 119) about the associational critique.

We have cited 1-4 references in line 34 and added the citation of Wolf & Broughton (2020).

2) Line 51-52: the statement about "first big-game hunters in the world" is embarrassingly wrong -- it ignores the abundant and much older record of mammoth-hunting (and horse-hunting, bison-hunting, and reindeer-hunting) in Eurasia.

We have removed this sentence

3) Line 73: replace "them self" with themselves (one word).

Changed

4) Lines 84-86: the sentence fits better in a Materials and Methods section, which should be moved to precede the Results section.

As journal style is for the Methods to be at the end of the main text, we would prefer to keep some overview of the methodology earlier in the manuscript to help guide readers.

5) Lines 116-120: this sentence should be in the Results section of the paper.

We have removed the following sentence: "Our results show a clear association between the temporal changes in density of all human sites, FPPs, and megafauna, as well as between the spatial distribution of the extinct megafaunal species and the FPP"

6) Lines 122-126: This should be placed in a Materials and Methods section.

As journal style is for the Methods to be at the end of the main text, we would prefer to keep some overview of the methodology earlier in the manuscript to help guide readers.

7) Lines 166-170: This should be placed in a Materials and Methods section preceding the Results.

As journal style is for the Methods to be at the end of the main text, we would prefer to keep some overview of the methodology earlier in the manuscript to help guide readers.

8) Lines 200-203: This should be placed in a Materials and Methods section preceding the Results.

As journal style is for the Methods to be at the end of the main text, we would prefer to keep some overview of the methodology earlier in the manuscript to help guide readers.

9) Fig. 5: the 'human scale' image makes the proboscidean silhouettes look relatively small. Were the fossil species actually that small?

Silhouettes were based on body size estimations from South Proboscideans which are smaller than north American taxa. So, this could suggest the figure is wrong but it is not.

10) Lines 222-224: This should be placed in the Materials and Methods section preceding the Results.

As journal style is for the Methods to be at the end of the main text, we would prefer to keep some overview of the methodology earlier in the manuscript to help guide readers.

11) Line 226: after "no-metric" insert (nm) (in parentheses).

"nm" was inserted after multidimensional scaling: nm-MDS, such is frequent in multivariate statistics.

12) Line 229: a word is missing between "Particularly," and "M. americanum".

The sentence was clarified

13) Line 239: the antecedent is unclear for the phrase "it is closer..."

The sentence was clarified

14) Line 250: Does the date '17,5 k years BP' refer to radiocarbon years?

In order to avoid ambiguities related to the reporting of dates, we now refer to calibrated dates as "cal BP" as suggested by Millard (2014:557).

15) Lines 425+: A Materials and Methods section usually precedes the analytical results and conclusions. Is this the journal's format?

Yes, this is the Journal's format

RESPONSE TO R2'S COMMENTS

I consider that the study you have made is novel and contributes to the debate around the LQE in South America. Even if I have been working in this field for quite a few years now, still surprises me how the debate around the causes of megafaunal extinction remains open and heated. In this context, every good contribution (such as this present one) is one more step to reach the definite answer (if possible).

I have several comments, suggestions and concerns about your manuscript that are detailed in the attached annotated PDF. I hope they are helpful and contribute to make your work better.

Best regards,

Natalia A. Villavicencio

Recommended changes R2 (made on the PDF):

1) comment below some taxonomic issues regarding two of the analyzed species. For the case of *Notiomastodon waringi* I suggest to change it to *Notiomastodon platensis*. For the case of *Lama gracilis*, I suggest to check in the references mentioned in the comment for those specimens that are no longer assigned to this species. Is necessary, a clarification about this taxonomic issues may be included in the supplementary materials.

The taxonomic status of *Lama gracilis* is unsolved. Specimens assigned to this taxon in the Pampa have been synonymized with *Vicugna vicugna* (Scherer 2009), while the remains from Kamac Mayu in Chile have been recently re-assigned to *Vicugna provicugna* (Labarca 2015).

-Scherer, Carolina Saldanha. "Os Camelidae Lamini (Mammalia, Artiodactyla) do plesistoceno da América do Sul: aspectos taxonômicos e filogenéticos." (2009).

-Labarca, Rafael. "La meso y megafauna terrestre extinta del Pleistoceno de Chile." *Publicación Ocas. Mus. Nac. Hist. Nat. Chile* 63 (2015): 401-465.

According to the most recent publications (see below), the validity of this species is questionable and it is considered a junior synonym of *Notiomastodon platensis*.

- D. Mothé, L.S. Avilla, M.A. Cozzuol, G.R. Winck, Taxonomic revision of the quaternary gomphotheres (Mammalia: Proboscidea: Gomphotheriidae) from the South America lowlands, *Quaternary International*, 276–277 (2012), pp. 2-7

- D. Mothé, M.P. Ferretti, L.S. Avilla The dance of tusks: rediscovery of lower incisors in the Pan-American proboscidean *Cuvieronius hyodon* revises incisor evolution in *Elephantimorpha* *PLoS One*, 11 (1) (2016), p. e0147009, 10.1371/journal.pone.0147009

- D. Mothé, L.S. Avilla Mythbusting evolutionary issues on South American Gomphotheriidae (Mammalia: Proboscidea) *Quaternary Science Reviews*, 110 (2015), pp. 23-35

We agree with the reviewer 2 regarding the taxonomy of South American proboscideans is still debated. Mothé et al. have proposed that *N. platensis* and *N. waringi* are the same species, but Prado & Alberdi 2015 (see also Recabarren et al. 2014) suggest that there are two different species or two geographic variants of a single one. Whether or not they are the same species, we prefer to consider both variants separately (as Prado & Alberdi) for a better modeling of the potential distributions. As the reviewer asked, we clarified this issue in the Legend of the Fig. 5.

We also agree with the reviewer regarding the status of *Lama gracilis* is unsolved and debated. Morphological and molecular arguments suggest that this species is probably the same than ***Vicugna vicugna*** or a vicariant of **it** in the low plains (Menegaz et al. 1989; **Scherer 2009**; Cajal et al. 2010; Weinstock et al. 2009; Cajal & Tonni 2010). Beyond the taxonomic status of this species we included it in the analysis because most of the literature still use “*L. graciis*” for extra-Andean records and (principally) because, in any case, it got extinct or retracted during the Final Pleistocene in South American low plains. As the reviewer asked, we clarified this issue in the Legend of the Fig. 3

2) Line 126. Replace site with sites.

It was replaced site with sites.

3) Line 131. References for these late records are needed

We have added the following reference which summarizes late records of megafauna: Politis, G. G., & Messineo, P. G. (2008). The Campo Laborde site: New evidence for the Holocene survival of Pleistocene megafauna in the Argentine Pampas. *Quaternary International*, 191(1), 98-114.

4) The time period (or moment) for which the models were built should be stated from here

We have replaced “considering species and FFP occurrences” with “considering species and FFP occurrences between 18k and 9k cal BP”

5) 170-171. This is already mentioned in the previous paragraph.

The sentence was removed

6) 238-239: Check grammar of this sentence

We have clarified the sentence.

7) 248: replace along with through

Done

8) 249: replace increasing with increase

Done

9) 250-252: I would rephrase this sentence since it is a bit confusing as it is. Something like: For southern South America, this pattern could be linked to

We have rephrased this sentence following an observation from another reviewer (see below).

“Although this pattern is probably linked to the expansion of large herbivore mammals and their predators during the end of the Late Pleistocene, it could be also influenced by taphonomic biases or the loss of fossil specimens over time”.

10) 256. decline of the radiocarbon dated fossil record. Not necessarily of the entire fossil record for this time period.

We have replaced “decline of the fossil record” with “decline of the radiocarbon dated fossil record”

11) 266: replace seems with seem

Done

12) Line 267: Not sure if the term species is correct here. Maybe density of megafaunal individuals, or just, density of megafauna, could be more appropriate.

We replaced “density of megafaunal species” with “density of megafauna”

13) Line 285: choose another word here, globally means: a nivel mundial, which is does not apply for this case

We have removed “globally”

14) Line 295-296: How does this fact affect your observation stated in lines 281-283? It is true that the spatial relation between the potential distribution of FPP and the potential areas of greater species richness may be indicative of a relation between both (hunting technology and presence of megafauna), but. what about the demography of humans at the time?. If you observe that human population is more broadly distributed through out the continent and not concentrated in the "hunting-areas", can you certainly make a statement of a relationship between changes in human population density and changes in megafaunal density due to hunting? Maybe you can suggest it, but not make any strong affirmation about it?.

The observation of the lines 281-283 refers to that: the human demography through time (not over space) is strongly related to the one of megafauna and FPP. Specifically, we mean that the increasing of the human density (observed in the SDP of humans) suddenly stops at the time of both megamammals and FPP curves started to decrease. The fact that human population is more broadly distributed throughout the continent and not concentrated in the "hunting-areas" do not

affect the previous statement because the stop of the increase of the human signal is a temporal (not a geographical) fact.

15) Line 304: reviewer suggest to replace located with focused

Done

16) Line 305: reviewer suggest to replace: "Pampean species and then to the Patagonian ones" with "Pampean species as well as to Patagonian ones"

Done

17) Line 308: Reviewer says: and possibly more abundant... An animal might be better adapted to a particular environment but not necessarily more abundant on it (due to competition with other species for example).

- We have replaced "where the large mammals were better adapted" with "where the large mammals were better adapted and possibly more abundant"

18) Line 311. Reviewer says: "Richness might be a tricky concept here. As it is, implies that more species of megafauna appeared on the continent at the time humans were arriving. If you are trying to state an increase in species richness at some particular areas in the continent (for example Pampas and Patagonia), you need to be more specific".

- We agree, we have removed the word "richness" so we only refer to abundance which is more accurate.

19) Line 313. Reviewer suggest to remove "from"

Done

20) Line 326. Replace "explored" with "exploited"

Done

21) Line 329. Specify that is the carrying capacity for humans.

Done

22) Line 334-335. You say global, but your climatic references are for the southern hemisphere and South America, as well as the timing for your LGM (~18 kaBP). Maybe the word global should be changed for something referring to the continent.

We have removed the word "globally" so it become clearer that we refer to South America only.

23) Line 337. Much less significant south of the 40°S? are you sure? Check this. I think the ACR has a stronger signal south of the 40°S than in the north of the continent, were the signal of the YD is usually seen.

- We agree with the reviewer this statement is not accurate enough. We have re-written a couple of sentences here (see below) in order to make clearer that what we consider important regarding the ACR is the fact that vegetation in Southern South America did not seem to have changed abruptly after the ACR (around 13k BP). On the contrary, the retraction of the open grasslands (preferred environment of megafauna) seems to have occurred progressively some centuries later.

We replaced: "This was a postglacial cooling period, which was much less significant south of the 40° S (including Pampas and Patagonia) than in low and middle latitudes of South America^{23,44}, and was the time when the megafauna reaches the maximum growth in density (Fig. 8). After the ACR, a warming period started at ca. 13 k BP. This warming period coincides with a cooling period in North America (the Younger Dryas) and with the retraction in southern South America of the open grasslands –the preferred habitat of the megafauna^{7,22,44,53}."

With: "This was a postglacial cooling period, occurred at the same time that the megafauna reaches the maximum growth in density (Fig. 8) and coincides with North American Bølling–Allerød warm stage⁵³. After the ACR, a warming period started at ca. 13 k BP, which coincides with a cooling period in North America (the Younger Dryas)¹⁶. Terrestrial paleovegetation proxies do not suggest any abrupt environmental change in Southern South America at the end of the ACR, but persistent cold/cool and wet conditions until 11,8 BP^{54,55}. The open grasslands –the preferred habitat of the megafauna^{7,47,48} seem to have retracted and reached their current distribution after 12,4 - 11,5 k cal BP in different areas of Patagonia^{7,32,33,47,56}, and probably later in the Pampas⁵⁷. Additionally, Pampa and Patagonia suffered a significant reduction of the territory, mainly affecting open areas, due to the rise in sea level^{47,48}

24) Line 348 (Fig. 8 reading). You must specify clearly from where the data of climate changes presented on this graph is coming from. References needed here.

The graph is from Pedro et al. 2016. We added the reference in the legend of the Fig.

25) 353-354. I suggest to check this again. The ACR had an effect in Patagonia's climate, specially for Southern Patagonia. For the Pampas the effects of that particular climate event may be less strong, I agree. Just a quick google search gives a series of recent articles indicating, for example, glacial advances in Patagonia at the time of the ACR. I strongly suggest again to be specific about the references for the data of climate changes that is being showed in figure 8. Without a clear source for that data, I cannot make any informed comments on that.

- We also agree with the observation of the reviewer 2. As stated above, we have modified the text in order to focus on the not abrupt environmental changes at the end of the ACR. With this aim we have replaced: "First, the faint effects of the ACR climatic event in Pampa and Patagonia (south of 40°S, see Fig. 8) seem not so congruent with a sudden decline of megafauna at ca. 12,9 k years BP (Fig. 8)" with: "First, the gradual effects of the end of the ACR climatic event in Pampa and Patagonia (Fig. 8) seem not so congruent with a sudden decline of megafauna at ca. 12,9 k cal BP (Fig. 8)".

26) 362-363. This is the timing for the cooling period ACR. If you want to signal the end of the ACR (~13) as the beginning of a warming period, be more specific.

- We agree with the reviewer 2. We have replaced “Finally, whereas in North America the initial decline of the megafauna (13 k years BP) is contemporaneous with the beginnings of the post glacial cooling period (the Younger Dryas: 13 - 11,7 k years BP), in South America the initial decline of megafauna (12,9 k years BP) is contemporaneous with the beginnings of a warming period (ACR: 14,7 - 13 k years BP).”

With: “Finally, whereas in North America the initial decline of the megafauna (13 k cal BP) is contemporaneous with the beginning of the post glacial cooling period (the Younger Dryas: 13 - 11,7 k cal BP), in South America the initial decline of megafauna (12,9 k cal BP) is contemporaneous with the beginning of the warming period starting after the ACR, at ca. 13 k cal BP.”

27) 362-366. a warming period could also have affected megafauna.

- We compare the cases of North America and South America because if climatic change were the main driver of the extinctions (on a continental scale) we expect in both hemispheres the time of the abrupt retraction of megafauna to be associated with similar climatic events. In this case is not so. In North America this occurs at the beginning of a cooling event (HD) and in South America at the end of a cooling event (ACR). Even so, we agree with the reviewer regarding a warming period could also have affected megafauna but as a secondary agent. To make clearer this idea we have replaced:

“All this evidence suggests that although megafaunal species could be under high ecological stress, they did not become extinct until humans using FPP appeared, suggesting again that human agency was the central and determining direct factor driving megafaunal extinctions.”

With “Although this change could have affected megafauna and put species under high ecological stress, they did not become extinct until humans using fluted projectile points appeared, suggesting again that human agency was a determining factor driving megafaunal extinctions.

28) I think is important to consider the limitations of your analysis, specially with regards to that your are seeing the agency of human hunters in a group of species and not on the whole spectrum of taxa that were present during the late Pleistocene of South America. This goes more as a reminder that it may be too bold to search for a unique driver of extinction for South America, or to conclude that humans where the determinant factor for this major extinction event from an analysis that only considers some of the entities involved.

- We agree with the reviewer. We have moderated the importance of humans as driver of extinctions by replacing “All this evidence suggests that although megafaunal species could be under high ecological stress, they did not become extinct until humans using FPP appeared, suggesting again that human agency was the central and determining direct factor driving megafaunal extinctions.”

- With: "Although this change could have affected megafauna and put species under high ecological stress, they did not become extinct until humans using fluted projectile points appeared, suggesting again that human agency could have been a determining factor driving megafaunal extinctions.

29) 375: it has been considered incongruent with a Blitzkrieg ("rapid" as you stated) model of overkill, not with overkill by humans as a more general process nor with regards to "human driven extinction" which might include a broad sort of drivers, not only direct hunting. Be more specific.

- We have removed this sentence because it is not important for the discussion and (we agree with the reviewer) it is a quite confusing.

30) 377: correlation is not causation. Be careful to say that your results "clearly show". Maybe "suggest" may be more appropriate.

We have changed "clearly show" with "suggest"

31) 385: Right, but are thought to have had low reproductive rates. We are not certain about this solely from the fossil record

We clarified the sentence: "As well, based on body mass estimations is probable that most species had low reproductive rates⁷...."

32) 392: As.....such just it could be nice to see some examples enumerated here

Suggestion incorporated. We added two species studied by Pires et al, that could become extinct by the impact of indirect effects.

33) Dataset 2 References

All the references of the radiocarbon dates included in the Dataset 2 were incorporated.

RESPONSE TO R3'S COMMENTS

Based on the objections mentioned above, it is up to the Nature editors to accept this paper for publication.

In the following revision the numbers indicated the lines.

1) 20: I do not know if the in the abstract acronyms may be used. However, please explain before what “FPP” means.

- We have replaced FPP with “Fishtail projectile point” in the Abstract and we have made clearer the use of the FPP acronym in the Introduction of the paper: “Fishtail projectile points (henceforth referred to as FPP)”

2) 44: says: “..13 and 12 k years BP..”. Please, specify is they are calibrated years.

- In order to avoid ambiguities related to the reporting of dates, we now refer to all calibrates dates as “cal BP” as suggested by Millard (2014:557).

3) In the supplementary table regarding the list of sites with fishtail points, to the data recorded from Uruguay, in many of them, the authors refer Weitzel et al. (2018) paper as the main reference. However, the primary source (Nami 2013) of those findings is not properly provided. I suggest citing the primary source from that data deserves to be cited. Actually, in that publication, tables are providing the precise origin, metric data, and other observations of each specimen. Also, the primary source to access the detailed data of most detailed published Uruguayan fishtail points are given in several papers published by Nami. All of them provide very much detail for the Uruguayan Fishtails points.

- We have replaced Weitzel (2018) with Nami (2013).

4) A similar situation occurs in the supplementary table of radiocarbon dates. Indeed, when it provides the dates from Cueva del Medio depicted in rows 22 to 28 whose laboratory identification is NUTA and Gr-N, the authors provides as main reference the Martin et al. (2015) and does not cite the primary source of the dating, which is Nami and Nakamura (1995). Curiously, this original article is only referred to in one of the Cueva del Medio dates obtained in the same Japanese laboratory. The dates processed in the Groningen lab are provided in Nami and Nakamura (1995), and if the authors wish a more recent reference to use, the totality of the dates for the Fishtail occupation are available in Nami (2019). They are also referred to in other articles (Nami 2007, and 2017).

Nami, H. G. 2007. Research in the Middle Negro River Basin (Uruguay) and the Paleoindian Occupation of the Southern Cone. *Current Anthropology*, 48, 164-176. <https://doi.org/10.1086/510465>

Nami, H. G. 2013. Archaeology, Paleoindian Research and Lithic Technology in the Middle Negro River, Central Uruguay. *Arch. Disc.*, 1, 1-22.

Nami, H. G. 2017. Silcrete as a valuable resource for stone tool manufacture and its use by Paleo-American hunter–gatherers in southeastern South America. *Journal of Archaeological Science: Reports*, 15, 539-560. <http://dx.doi.org/10.1016/j.jasrep.2016.05.003>

Nami, H. G. 2019. Paleoamerican Occupation, Stone Tools from the Cueva del Medio, and Considerations for the Late Pleistocene Archaeology in Southern South America. *Quaternary*, 2, 28. Special Issue “Changing Nature of Socio-Ecological Interactions in the Americas: From

PalaeoAmerican through to Present Day". doi:10.3390/quat2030028

Nami, H.G., Nakamura, T. Cronologia radiocarbonica con AMS sobre muestras de huesos procedentes del sitio Cueva del Medio (Ultima Esperanza, Chile). An. Inst. Pat. Ser. Cs. Soc. 1995, 23, 125-133.

- We have replaced Martin et al. (2018) with Nami and Nakamura (1995) in all the references of the dates (NUTA and Gr-N) from Cueva del Medio.

RESPONSE TO R4'S COMMENTS

This paper should be published with minimal revisions. The authors show a clear correlation between increasing human and declining megafaunal populations in the Late Pleistocene of South America. This is an important result and one that deserves publication. Specific comments follow:

1) Lines 32-33 – the authors state that “there is a general agreement that the combined role of human activity – particularly hunting – and climate change related to late glacial fluctuations were the major drivers of extinctions on a global scale.” I am skeptical of this. Some people would argue that only human hunting mattered. Others would argue for the role of hyperdisease, habitat modification, or even extraterrestrial impact. I think it would be better to state that cause continues to be debated.

- We agree. We have replaced the sentence “However, there is a general agreement that the combined role of human activity – particularly hunting – and climate change related to late glacial fluctuations were the major drivers of extinctions on a global scale” with “Some people argue that only human hunting mattered, others argue for the role of climate change, hyperdisease, habitat modification, or even extraterrestrial impact” and added two new references (see below) to cover with citations these different opinions.

Rothschild, B. M., & Laub, R. (2006). Hyperdisease in the late Pleistocene: validation of an early 20th century hypothesis. *Naturwissenschaften*, 93(11), 557-564.

Firestone, R. B., West, A., Kennett, J. P., Becker, L., Bunch, T. E., Revay, Z. S., ... & Dickenson, O. J. (2007). Evidence for an extraterrestrial impact 12,900 years ago that contributed to the megafaunal extinctions and the Younger Dryas cooling. *Proceedings of the National Academy of Sciences*, 104(41), 16016-16021.

2) Lines 67-69 “Based on this lack of evidence, along with the general agreement regarding humans reaching South America in “pre-Clovis” times”. Again, all archaeologists do not agree with this sentiment. In fact, if one considers the possibility that South America had no human population until the appearance of fishtail points, the patterns identified in this paper make a

lot more sense. As an example of a recent Clovis-first study, see for example: Fiedel, S.J. (2017) The Anzick genome proves Clovis is first, after all. *Quaternary International* 444, 4–9. Also, the Surovell et al. paper cited (21) makes a similar argument.

We agree with the Reviewer 4. In the sentence referred by the reviewer, we wanted to mean that “south American archaeologist” agree regarding humans reaching South America in “pre-Clovis” times”. To make this clearer, we have change the sentence as follow: “Based on this lack of evidence, and assuming that humans reached South America in “pre-Clovis” times, South American archaeologists.....”.

Regarding the second part of the Reviewer’s comment, we have added a line (see highlighted in yellow below) that we feel responds to the observation of the reviewer: “The tight fit between the behavior of both late Pleistocene phenomena –FPP and human population expansion– (Fig. 1) may be indicative that rapid and initially successful dispersal of FPP technology drove the quick population growth of the earliest foragers or, from a more conservative perspective, that FPP were the first colonizers of South America.

3) Lines 92-95, I recommend stating this sentence differently. This sentence forms the basis of hypothesis testing. It could be better stated as, “If humans caused the extinctions, we expect to see an inverse correlation between human and megafaunal populations in time and a positive correlation in space.”

We have made the change suggested by the Reviewer 4, which makes much clearer the principal testing hypothesis of the paper.

4) Lines 108-109 – “If megafauna were a central resource in human economy, we expect that changes in megafaunal density and distribution must have impacted human population dynamics.” I would expect the opposite, wouldn’t human populations impact megafaunal population dynamics?

We agree with reviewer 4. In fact, we would expect both: that human population affects megafauna (as suggested by Reviewer 4) and also that megafauna density affects human population (as we said in lines 108-109). On this basis we have changed the sentence as follow: If megafauna were a central resource in human economy, we not only expect that human impact megafaunal population dynamics, but also that changes in megafaunal density and distribution impact human population.

5) Lines 126-130. Here it suggested based on an SPD that megafauna populations increase through time from ca. 18,000 to 13,000 BP. This pattern, however, has been repeatedly observed in other contexts and is almost certainly caused by taphonomic bias, or the loss of fossil specimens over time. See the following:

▫Bluhm, Lara E., and Todd A. Surovell. “Validation of a Global Model of Taphonomic Bias Using Geologic Radiocarbon Ages.” *Quaternary Research*, 2018.

▫Surovell, T. A., and Spencer R. Pelton. “Spatio-Temporal Variation in the Preservation of Ancient Faunal Remains.” *Biology Letters* 12 (2016): 20150823.

▫Surovell, Todd A., and P. Jeffrey Brantingham. “A Note on the Use of Temporal Frequency Distributions in Studies of Prehistoric Demography.” *Journal of Archaeological Science* 34 (2007): 1868–77.

▫Surovell, Todd A., Judson Byrd Finely, Geoffrey M. Smith, P. Jeffrey Brantingham, and Robert L. Kelly. "Correcting Temporal Frequency Distributions for Taphonomic Bias." *Journal of Archaeological Science* 36 (2009): 1715–24.

- Surovell and Pelton (2016) have specifically developed a taphonomic correction curve for South American Pleistocene fauna. I would not insist that the author use it, but they should at least acknowledge that the apparent growth in the number of dated specimens might have nothing to do with population dynamics but is likely instead just preservation or taphonomic bias.

We agree with the Reviewer 4. We have not used the correction curve developed by Surovell & Pelton (2016), but we discussed the topic. We have not modified the lines 126-130 (Results) because there we only describe the results of the SPD. Nevertheless, we have modified the sentence referring to this point in the discussion (see below) in order to make clear that taphonomic biases could be also a determining factor of the observed pattern. As the reviewer recommended we have also added a couple of the new references.

"Although this pattern is probably linked to the expansion of large herbivore mammals and their predators during the end of the Late Pleistocene, it could be also influenced by taphonomic biases or the loss of fossil specimens over time. If the expansion was real, it could be related to the favorable environmental condition for herbivores after the end of the Last Glacial Maximum^{47,48}, and especially in the Pampas and Patagonia regions"

5) Lines 254-258: "Although this pattern could be linked to the expansion of large herbivore mammals and their predators during the end of the Late Pleistocene²², it could be also influenced by taphonomic biases or the loss of fossil specimens over time (Surovell & Pelton 2016; Bluhm et al. 2019). If the expansion was real, it could be related to the favorable environmental condition for herbivores after the end of the Last Glacial Maximum".

Bluhm, L. E., & Surovell, T. A. (2019). Validation of a global model of taphonomic bias using geologic radiocarbon ages. *Quaternary Research*, 91(1), 325-328.

- **The same issue should be acknowledged again in lines 248-252.**

Ok, as we have mentioned above, we have modified the sentence considering the observation of the reviewer 4.

7) Lines 268-271 - "As shown in Fig. 1, during the initial period of peopling, between 15 and 13 k years BP, the archaeological signal (and probably human population density stayed extremely low, until an irruptive growth dynamic occurred just after the spread over southern South America." I would suggest that the authors at least consider the possibility that preClovis in South America is not real. This would provide a much simpler explanation for why archaeological sites are so rare and why megafaunal populations are unaffected.

We agree with the reviewer and we included this observation in the new version of the manuscript (see below).

"Although a simpler explanation for why archaeological sites are so rare and why megafaunal

populations are unaffected until 12,9 k cal BP could be that pre-Clovis in South America is not real, this could be also a result of that earliest human dispersal was associated with broad spectrum foraging, and that human population density was too low to affect expanding large mammals during the earlier dispersal times”

Reviewers' Comments:

Reviewer #2:

None

Reviewer #4:

Remarks to the Author:

The authors addressed my comments, and this paper should be published as is. It is an important contribution to the literature on Pleistocene extinctions in the Americas.